# Oral Cladribine Impairs Intermediate, but Not Conventional, Monocyte Transmigration in Multiple Sclerosis Patients across a Model Blood-Brain Barrier

**DOI:** 10.3390/ijms24076487

**Published:** 2023-03-30

**Authors:** Linda Y. Lin, Pierre Juillard, Simon Hawke, Felix Marsh-Wakefield, Georges E. Grau

**Affiliations:** 1Vascular Immunology Unit, School of Medical Sciences, Faculty of Medicine and Health, The University of Sydney, Sydney, NSW 2006, Australia; 2Central West Neurology and Neurosurgery, Orange, NSW 2800, Australia; 3Liver Injury and Cancer Program, Centenary Institute, Sydney, NSW 2006, Australia; 4Human Cancer and Viral Immunology Laboratory, The University of Sydney, Sydney, NSW 2006, Australia

**Keywords:** multiple sclerosis, blood-brain barrier, cladribine, monocytes, dendritic cells, spectral flow cytometry, trans-endothelial migration

## Abstract

Multiple sclerosis (MS) is a disease in which the immune system damages components of the central nervous system (CNS), leading to the destruction of myelin and the formation of demyelinating plaques. This often occurs in episodic “attacks” precipitated by the transmigration of leukocytes across the blood-brain barrier (BBB), and repeated episodes of demyelination lead to substantial losses of axons within and removed from plaques, ultimately leading to progressive neurological dysfunction. Within leukocyte populations, macrophages and T and B lymphocytes are the predominant effectors. Among current immunotherapies, oral cladribine’s impact on lymphocytes is well characterised, but little is known about its impact on other leukocytes such as monocytes and dendritic cells (DCs). The aim of this study was to determine the transmigratory ability of monocyte and DC subsets in healthy subjects and untreated and cladribine-treated relapse-remitting MS (RRMS) patients using a well-characterised model of the BBB. Peripheral blood mononuclear cells from subjects were added to an in vitro transmigration assay to assess cell migration. Our findings show that while prior treatment with oral cladribine inhibits the migration of intermediate monocytes, it has no impact on the transmigration of DC subsets. Overall, our data indicate a previously unrecognised role of cladribine on intermediate monocytes, known to accumulate in the brain active MS lesions.

## 1. Introduction

Myelin is essential for facilitating the rapid transmission of electrical impulses along axons [1]. Myelinated axons are predominantly found in white matter, connecting neurons in different regions of the brain to allow for information exchange and communication [1]. The hallmark of MS is the focal autoimmune destruction of white matter in the central nervous system (CNS), termed demyelinating plaques. These plaques are characterised by a substantial loss of axons, resulting in reduced electrical impulses that manifest as symptoms of neurological dysfunction [2] and cognitive disability [3]. Whilst the aetiology of MS is unknown, its pathogenesis is hypothesised to involve immune, environmental and genetic factors [1]. One crucial feature of MS pathogenesis is the transmigration of leukocytes to the perivascular spaces across the blood-brain barrier (BBB) [4,5,6]. The BBB prevents the entry of toxins, pathogens and neurotransmitters into the brain that can damage neurons [7]. The autoreactive nature of MS exposes endothelial cells to pro-inflammatory cytokines such as interferon-gamma (IFN-g) and tumour necrosis factor (TNF) which disrupt the BBB by redistributing junctional protein expression [7]. Ultimately, BBB disruption permits leukocyte transmigration, further increasing protease secretion and damage to tight endothelial junctions. The key role of the immune system in the pathogenesis of MS is exemplified by the success of multiple drugs that target various components of the immune system and their migration across the BBB [8]. Nonetheless, despite their effectiveness in reducing CNS inflammation, preventing disease progression remains an issue for some MS patients. 

Adaptive immune cells are the dominant cells involved in MS pathogenesis, causing local inflammation and demyelination through the secretion of cytokines, chemokines, or antibodies [9]. As lymphocytes have a crucial role in MS pathogenesis, several studies have utilised an in vitro BBB model, based on the Boyden chamber assay, to investigate their phenotype and migratory behaviour. Using cytokine-stimulated endothelial cells, the pro-inflammatory conditions of the MS brain can be effectively mimicked [10,11]. However, the role of innate immune cells, particularly monocytes and DCs in MS pathogenesis, has been overlooked. 

Monocytes and DCs have proinflammatory roles via the processing and presentation of nuclear and protein antigens from infectious agents that may induce cross-reactive adaptive responses to myelin [12]. This activation and production of pro-inflammatory cytokines potentially disrupts the BBB, resulting in transmigration, perpetuating inflammation and subsequent MS pathogenesis [13]. Immunohistochemical analysis revealed that monocytes are more abundant than lymphocytes in early MS lesions [14], suggesting that monocytes are the dominant cells in the early stage of plaque formation. Within the monocyte population, a significant increase in non-classical monocytes and a decrease in classical monocytes were reported, but their exact roles in MS pathogenesis remain unclear [15]. Similarly, DCs are abundant in MS lesions with altered functions compared to in healthy subjects: conventional DCs (cDCs) were found to accumulate in white matter plaques [16], while plasmacytoid DCs (pDCs) were observed in leptomeninges [17]. Additionally, immature DCs have been identified within the centre of lesions, whilst mature DCs were confined to the edges [18]. There are conflicting results regarding the circulating DC subsets in MS, and most studies have omitted the inclusion of a treatment group for comparison or have not investigated the transmigratory ability of monocytes and DCs [19]. Additionally, the impact of current MS therapeutics on monocytes and DCs remains to be investigated. 

Currently, numerous medications are available to manage the course of MS, including interferon beta (IFN-b), fingolimod and oral cladribine. IFN-b is the first disease-modifying therapy to treat MS, which increases the concentration of anti-inflammatory agents and downregulates the expression of proinflammatory cytokines to reduce migration of inflammatory cells across the BBB [20]. Fingolimod is classified as a sphingosine l-phosphate receptor modulator used to treat MS. It binds to sphingosine 1-phosphate receptors with high affinity, preventing lymphocyte egress from lymphoid tissues into the blood and CNS, where they can cause inflammation [21]. However, there are limited studies on the effects of these medications on innate immune cells, particularly monocytes and DCs. Oral cladribine is one of two “immune re-set” drugs used in the treatment of RRMS, referring to a small subset of drugs that induce immunomodulatory effects for long after they have been eliminated, potentially conferring immune tolerance to the CNS. Cladribine was globally approved in 2017 [22] and is activated by deoxycytidine kinase (DCK) and incorporated into DNA which inhibits synthesis and repair, resulting in apoptosis [23]. Lymphocytes have high DCK content resulting in high levels of apoptosis induced by oral cladribine as reported by proliferation and apoptosis assays performed on peripheral blood mononuclear cells (PBMCs) [24]. This reduces the expression of pro-inflammatory cytokines and adhesion molecules, restricting the transmigration of immune cells [25]. Oral cladribine is administered intermittently in short courses, 12 months apart, demonstrating a decrease in both T and B lymphocyte counts [26]. The mechanisms underlying its immunomodulatory effect are under intense investigation, and its effect on monocytes and DCs remains largely unknown.

An important pathogenic step in MS is the transmigration of leukocytes, with various subsets extensively studied via an in vitro BBB model [27,28,29]. However, the phenotype and transmigratory ability of innate immune cells remains elusive. By improving our understanding of innate immune cells in MS, new mechanisms of action of cladribine may be revealed. In this study, cladribine altered the transmigration of intermediate monocytes but had no significant impact on the migratory patterns of other monocyte and DC subsets. This is important, because it highlights the significance of intermediate monocytes in MS pathogenesis and demonstrates the effect of cladribine on monocytes. 

## 2. Results

### 2.1. Oral Cladribine Has No Effect on Circulating Monocyte or DC Levels

Subsets of monocytes were differentiated using CD14 and CD16 from the single cell gating (Appendix A), defining classical (CD14^+^CD16^−^), intermediate (CD14^+^CD16^+^) and non-classical monocytes (CD14^dim^CD16^+^). The overall DCs population was identified by the lack of CD14 and CD16 expression, which was subdivided by the levels of CD38 and CD45RA expression. This corresponded to mature DCs (CD38^+^CD45RA^−^) and immature pDCs (CD38^−^CD45RA^+^). CD38^−^CD45RA^−^ cells were further gated on CD197 expression, resulting in CD38^−^CD45RA^−^CD197^+^ DCs. (Figure 1).

Fresh PMBCs were obtained from healthy subjects and untreated and oral-cladribine-treated RRMS patients and counted before being added to the transmigration assay. Individual values for each patient were presented in a scatter plot with the median shown (Figure 2). The total PBMC numbers were significantly lower in oral-cladribine-treated than in untreated RRMS patients even months after the treatment (Figure 2A). However, the numbers of total monocytes and DCs did not differ between groups (Figure 2B). 

The absolute numbers of intermediate monocytes in untreated RRMS patients were significantly higher than in healthy subjects. However, the numbers of intermediate monocytes were not significantly greater in untreated RRMS patients compared to in oral-cladribine-treated RRMS patients. Additionally, there was no significant difference in the absolute numbers of classical, non-classical monocyte, immature and mature pDCs and CD38^−^CD45RA^−^CD197^+^ DCs among the three study groups (Figure 2C,D).

### 2.2. DC Subset Transmigration Is Unaltered following Cladribine

Total DCs did not migrate, and no significant differences in the absolute numbers of migrated DCs were observed across all groups (Figure 3A). Additionally, no definitive transmigratory pattern of CD38^+^CD45RA^+^ DCs in healthy subjects and untreated RRMS patients was identified (Figure 3B). The numbers of migrated CD38^+^CD45RA^+^ DCs were similar across the three groups (Figure 3B). Similarly, CD38^+^CD45RA^−^ DCs did not migrate across groups (Figure 3C), with no difference in migrated numbers (Figure 3C). CD38^−^CD45RA^+^ DCs did not show a migratory pattern in any group (Figure 3D), and the total numbers of migrated CD38^−^CD45RA^+^ DCs were similar in the three groups (Figure 3D). Likewise, there was no migration of CD38^−^CD45RA^−^CD197^+^ DCs in all groups, but healthy subjects had significantly lower numbers of migrated than non-migrated cells (Figure 3E). However, the absolute numbers of migrated cells in this subset tended to be lower in cladribine-treated than in untreated RRMS patients and healthy subjects, without reaching significance (Figure 3E).

### 2.3. Oral Cladribine Inhibits the Transmigration of Intermediate Monocytes

Total monocytes actively migrated in healthy subjects, but not in untreated and oral-cladribine-treated RRMS patients (Figure 4A). There was no significant difference between the absolute numbers of migrated monocytes in healthy subjects and untreated and cladribine-treated RRMS patients (Figure 4A). Intermediate monocytes significantly migrated in both healthy subjects and untreated RRMS patients. In contrast, intermediate monocytes lost their active migration in cladribine-treated RRMS patients (Figure 4B). Absolute numbers of migrated intermediate monocytes were lower in cladribine-treated RRMS patients compared to in untreated RRMS patients (Figure 4B). 

Classical monocytes similarly migrated across groups (Figure 4C). There was no difference in the number of migrated classical monocytes across groups (Figure 4C). Contrary to classical monocytes, there was no migration of non-classical monocytes in any of the three groups (Figure 4D). The numbers of migrated non-classical monocytes were lower in oral-cladribine-treated RRMS patients compared to in untreated RRMS patients (Figure 4D).

## 3. Discussion

This study in RRMS patients identifies previously unrecognised transmigratory ability of monocytes and DCs subsets using an in vitro BBB model. We also demonstrated for the first time the impact of oral cladribine on the transmigration of monocytes and DCs. 

To determine the effect of oral cladribine on circulating innate immune cells, fresh PBMC were analysed prior to transmigration. Our study revealed no significant differences between the absolute circulating numbers of monocytes and DCs across groups. Similarly, no differences in absolute monocyte numbers were reported after two to four years of oral cladribine treatment [30]. Oral cladribine significantly reduced the numbers of monocytes in 24 h and DCs in 72 h [31]. These differences highlight the fast action of cladribine on monocytes and DCs, as depleted cells can be repopulated in months [32]. 

Transmigration of intermediate monocytes was impaired in oral-cladribine-treated RRMS patients. Intermediate monocytes exhibit a pro-inflammatory phenotype due to increased expression of toll-like receptors (TLR) such as TLR2, TLR4 and TLR5, compared to classical and non-classical monocytes [33]. CD80, CD86 and HLA-DR are also highly expressed, indicating that intermediate monocytes are potent APCs [31]. Under physiological conditions, intermediate monocytes have access to CSF-filled spaces to interact with autoreactive T cells that can initiate MS [34]. However, in RRMS patients, CCR5 expression is upregulated on intermediate monocytes, and their corresponding ligands are highly expressed in inflammatory lesions of MS, resulting in mass infiltration into the perivascular space [35]. Furthermore, mutation of the 32-base pair deletion allele of CCR5, CCR5Δ32, leads to slower disease progression due to diminished migration of leukocytes, thereby reducing inflammation and damage to myelin [36]. This suggests a possible detrimental role of intermediate monocytes in MS pathogenesis, propagated by CCR5 binding. Additionally, cladribine has been demonstrated to significantly reduce CCR5 expression in T cells and monocytes as part of its therapeutic action [37], supporting the notion of detrimental intermediate monocytes.

Classical monocytes preferentially transmigrated across the stimulated BBB in all groups. This supports the current literature depicting abundant and rapid migration of classical monocytes due to high expression of matrix metalloproteinases (MMP)-2 and MMP-14 expression [38,39]. In contrast to classical monocytes, non-classical monocytes did not preferentially transmigrate across groups, supporting the literature [35]. The adherence of non-classical monocytes to the inflamed endothelium causes further damage to the BBB, promoting self-migration and T cell migration into the parenchyma [32]. Further investigations may include cytokines such as CXCL1 in the transmigration assay to more closely mimic the BBB [38]. 

CD38^−^CD45RA^+^ and CD38^+^ CD45RA^+^ DC are subsets of pDC, due to CD45RA expression [40]. CD38 can then be used to differentiate between immature (CD38^−^) and mature (CD38^+^) pDC [41]. Our results show no active migration of immature or mature pDCs in healthy subjects, reflecting the immune-privileged environment of the CNS [42]. Another subset of immature DCs, CD38^−^CD45RA^−^CD197^+^ DCs, did not exhibit migratory capacity across the BBB. In MS, an abundance of immature pDCs is found in demyelinating lesions of RRMS patients where maturation occurs after transmigration across the BBB [43]. This suggests that DCs migration is enhanced by chemokines, such as CCL2, CCL5 and CCL19 which are elevated in MS lesions [44]. This is evident in our results, that is, no migration was observed in untreated patients, as our transmigration assay did not contain any chemokines.

One strength of this study is the utilisation of the transmigration assay which is a widely accepted in vitro BBB model. It is a well-calibrated assay that allows for the selective migration of immune cells [45,46,47]. However, in this model, only endothelial cells are used. The inflammatory brain environment is difficult to accurately replicate, as the BBB consists of other components such as glial limitans and astrocyte foot processes, all contributing to selective migration. Furthermore, our study did not address chemokine effects on leukocyte transmigration. Hence, future experiments should consider the use of a triple culture in vitro model (endothelial cells, astrocytes and pericytes) with chemokines such as CCL5, CXCL1, CCL19 and CCL21 that have shown to enhance the migration of monocytes [40] and DCs [48]. Increasing the sample size of the cladribine-treated MS cohort (*n* = 7) would also strengthen future studies.

In summary, this study investigated the transmigratory ability of monocytes and DC subsets in a BBB model. The migratory abilities of three monocyte subsets and five DC subsets were assessed, with oral cladribine specifically inhibiting the migration of intermediate monocytes. In contrast, oral cladribine had no impact on the migratory pattern of DCs. These data suggest that an important mechanism of action of oral cladribine is to inhibit the translocation of intermediate monocytes into the CNS, a leukocyte subset known to hold a crucial role in the pathogenesis of active MS lesions. It will be interesting to assess for how long this effect lasts and to determine what molecules and cellular systems induce it. Taken together, our data reinforce oral cladribine as a monocyte re-setting medication, exerting an effect for long after the drug has been eliminated. Overall, this strengthens our current understanding of MS pathogenesis and provides greater insights into oral cladribine’s mechanism of action.

## 4. Methods and Materials

### 4.1. Patient Characteristics

RRMS was diagnosed using the McDonald 2017 criteria [49] in patients from neurology clinics at Central West Neurology and Neurosurgery in Orange, NSW and the Brain and Mind Centre at the University of Sydney. Ethical consent for the study was received from the University of Sydney Research Integrity and Ethics Administration (Project No: 2018/377). 

Blood samples were taken from healthy subjects (*n* = 20) and untreated (*n* = 12) and cladribine-treated RRMS patients (*n* = 7) with consent (Appendix A). Healthy controls were not excluded, if they had a family history of neurological diseases. Untreated RRMS patients were classified as having active disease accompanied by new neurological symptoms and new T2 FLAIR MRI lesions within six months of blood draw. RRMS patients were not treated with steroids. For cladribine-treated RRMS patients, blood was drawn four months after starting treatment (Mavenclad^(R)^; five consecutive days at 1.75 mg/kg/day orally, repeated one month later). 

### 4.2. Isolation of PBMCs from Whole Blood

Fresh whole blood from healthy subjects and untreated and cladribine-treated RRMS patients was collected in EDTA tubes at room temperature (20–24 °C) and diluted with an equal volume of phosphate-buffered saline (PBS) without calcium and magnesium (#D8537; Sigma-Aldrich, St. Louis, MO, USA). To isolate PBMCs, 15 mL of Ficoll-Paque (#171440-02 GE Healthcare Pharmacia, Uppsala, Sweden) was overlaid with 30 mL of diluted blood and centrifuged at 400× *g* for 30 min at 20 °C with the brakes off.

PBMCs were collected from the plasma/Ficoll interface, diluted in PBS-2% FBS, centrifuged twice at 120× *g* for 10 min at room temperature and resuspended in PBS-2% FBS. PBMCs were enumerated with a Countess II Automatic Cell Counter (ThermoFisher Scientific, North Ryde, NSW, Australia) and stained with Trypan Blue to determine the number of viable cells before being added to the transmigration assay. 

From each patient, three different cell samples were collected: A fresh cell sample consisting of PBMCs after immediate isolation from whole blood;Migrated cells consisting of PBMCs in the lower chamber of the transmigration assay;Non-migrated cells consisting of PBMC in the upper chamber of the transmigration assay.

To obtain the absolute numbers of non-migrated cells, their numbers were calculated by subtracting the number of migrated PBMC from the total PBMC added to the chamber. This took into consideration the cells that were trapped between the stimulated brain endothelial monolayer. 

### 4.3. Transmigration Assay—An In Vitro Model of the Human BBB

The transmigration assay was developed according to the Boyden^(R)^ chamber protocol described in previous studies [27,28,29]. The assay consisted of an in vitro BBB system with a transwell insert, separated into two compartments: the upper and lower chambers. The transwell insert was a polycarbonate membrane with 3 µm pores (#3414 Costar Pharma, Smithfield, Australia) that was coated with human cerebral microvascular endothelial cells grown to confluence (hCMEC/D3, kind gift from Pierre-Olivier Couraud, Institut Cochin, Université René Descartes, Paris, France) modelling the physiological fidelity of the BBB. HCMEC/D3 cells were stimulated with pro-inflammatory cytokines—IFN-g and TNF—to mimic the inflammatory conditions of an MS patient’s brain vessels. 

On day 0, rat tail collagen type I (#C3867 Sigma-Aldrich, St. Louis, MO, USA) was diluted with sterile Baxter water (#A4217 Baxter, Old Toongabbie, Australia) and filtered to produce a 3% collagen solution to coat the transwell insert. Each transwell insert was incubated at 37 °C for 45 min for an optimal endothelial cell attachment, before the collagen solution was discarded. A transparent monolayer of collagen remained on each transwell insert. A volume of 1.5 mL of hCMEC/D3 (300,000 hCMEC/mL) was added to the collagen-coated transwell and settled for 20 min to maximise the cell attachment. A volume of 2.6 mL of warmed 1% antibiotic antimycotic (AA) solution (1:100 dilution; #A5955 Sigma-Aldrich, St. Louis, MO, USA) in a complete EBM-2 medium with 5% FBS (Sigma-Aldrich, St. Louis, MO, USA) was then added to the lower chamber and incubated overnight at 5% CO_2_ and 37 °C. 

On day 1, the medium was discarded, and each insert was placed in a new 6-well plate containing 2.6 mL of warm 1% AA in complete EBM-2, before another 1.5 mL of EBM-2 was added for overnight incubation at 5% CO_2_ and 37 °C. This was repeated on day 2. 

On day 3, each insert was placed in a new 6-well plate containing 2.6 mL of warm 1% AA in complete EBM-2 supplemented with TNF (5 ng/mL, Peprotech, Rocky Hill, NJ, USA) and IFN-g (10 ng/mL, Peprotech, Rocky Hill, NJ, USA), before 1.5 mL of EBM-2 supplemented with TNF (5 ng/mL, Peprotech, Rocky Hill, NJ, USA) and IFN-g (10 ng/mL, Peprotech, Rocky Hill, NJ, USA) was added and then incubated at 5% CO_2_ and 37 °C. 

On day 4, the confluence and integrity of hCMEC/D3 monolayer were assessed at least 3 h prior to performing the transmigration assay using Alexa Fluor^TM^ 555-conjugated wheat germ agglutinin (WGA) (#W32464 Invitrogen; ThermoFisher Scientific, NJ, USA). WGA stained the hCMEC/D3 monolayer by binding to N-acetylglucosamine and N-acetylneuraminic acid residues [50]. 

Subsequently, each transwell insert was placed in a new 6-well plate containing 2.6 mL of warm WGA (5 μg/mL), before 1 mL WGA was added to the insert. The plate was incubated for 10 min in the dark at room temperature and then washed twice with PBS twice before adding 1 mL of warm 1% AA in complete EBM-2. Confluence was assessed using a fluorescence microscope, and inserts were washed with PBS and transferred to a new 6-well plate filled with 2.6 mL of warmed EBM-2. 

To prepare for transmigration, each insert was transferred to a new 6-well plate containing 2.6 mL of warm 3% FBS in an RPMI 1640 medium (#R8758 Sigma-Aldrich, St. Louis, MO, USA). Then, 2.34 × 10^6^ isolated PBMCs (1.56 × 10^6^/mL) resuspended in RPMI-3% FBS were added to the upper chamber. The transwell inserts were incubated for 14–18 h at 5% CO_2_ and 37 °C. 

Collected cells were stained with Trypan Blue and counted with a Countess II Automatic Cell Counter to determine the number of viable cells before flow cytometry.

### 4.4. Spectral Flow Cytometry

Flow cytometry staining was conducted immediately after PBMC isolation in the fresh sample. Non-migrated and migrated cells were stained on the day of completion of the transmigration assay. PMBCs were centrifuged at 500× *g* for 5 min at room temperature to resuspend the pellet in an FACS buffer (PBS supplemented with 0.5% BSA, #A7906-100G Sigma-Aldrich, St. Louis, MO, USA) and 2 mM EDTA (#0105-500G, AMRESCO Inc., Solon, OH, USA). PBMCs were then incubated with a staining mix containing an FACS buffer and fluorescently conjugated antibodies (Appendix A) for 20 min at 4 °C, washed twice and fixed in 4% paraformaldehyde for 20 min at room temperature. Data were acquired on an Aurora spectral flow cytometer (Cytek^®^, Fremont, CA, USA) and analysed with FlowJo software v10.7 (BD Biosciences, Ashland, OR, USA). 

### 4.5. Statistical Analysis

Statistical analyses were performed with Prism 9.0.2 GraphPad software (San Diego, CA, USA). A Wilcoxon matched-pairs signed-rank test was used to compare paired migrated and non-migrated cell samples in study groups. A Kruskal–Wallis nonparametric one-way ANOVA with Dunn’s multiple comparisons test was used to compare values across study groups. A *p*-value of ≤0.05 was considered significant.

## Figures and Tables

**Figure 1 ijms-24-06487-f001:**
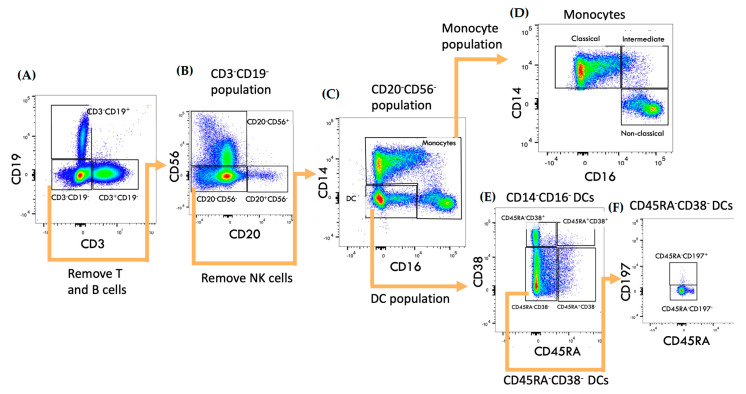
Gating strategy of monocytes and DC subsets. (**A**,**B**) Lymphocytes were first removed by selecting on CD3^−^CD19^−^ gate, which was then gated on CD56 and CD20. (**C**) Separation of mono–cytes and DCs. (**D**) Classical (CD14^+^CD16^−^), intermediate (CD14^+^CD16^+^) and non–classical (CD14^dim^CD16^+^) monocytes. (**E**) DCs subsets classification. (**F**) Identification of CD38^−^CD45RA^−^CD197^+^ DCs. The sequence of gating is indicated by the arrows.

**Figure 2 ijms-24-06487-f002:**
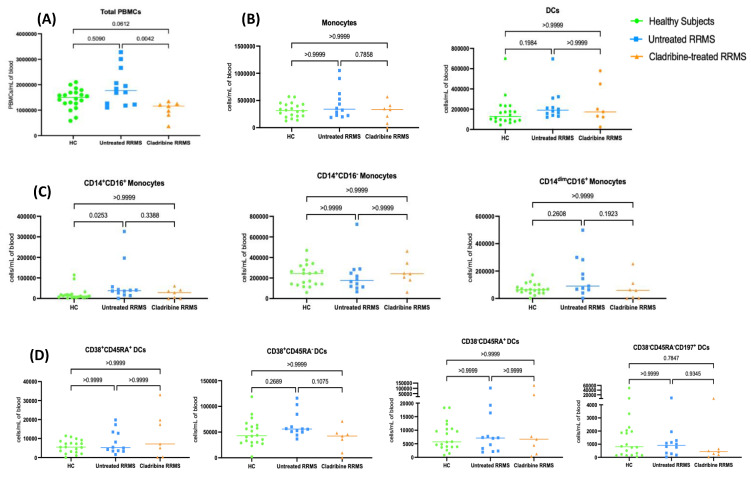
Cladribine has no effect on circulating monocyte or DC levels. Fresh PBMCs were immediately collected and phenotyped from healthy subjects (*n* = 20), untreated RRMS patients (*n* = 12) and cladribine–treated RRMS patients (*n* = 7). (**A**) Total fresh PBMCs/mL of blood. (**B**) Total numbers of monocytes and DCs per millilitre. (**C**) Total numbers of monocyte subsets per millilitre. (**D**) Total numbers of DCs subsets per millilitre. Analyses were conducted with Kruskal–Wallis nonparametric one–way ANOVA with Dunn’s multiple comparisons test. The significance was dictated by *p* < 0.05.

**Figure 3 ijms-24-06487-f003:**
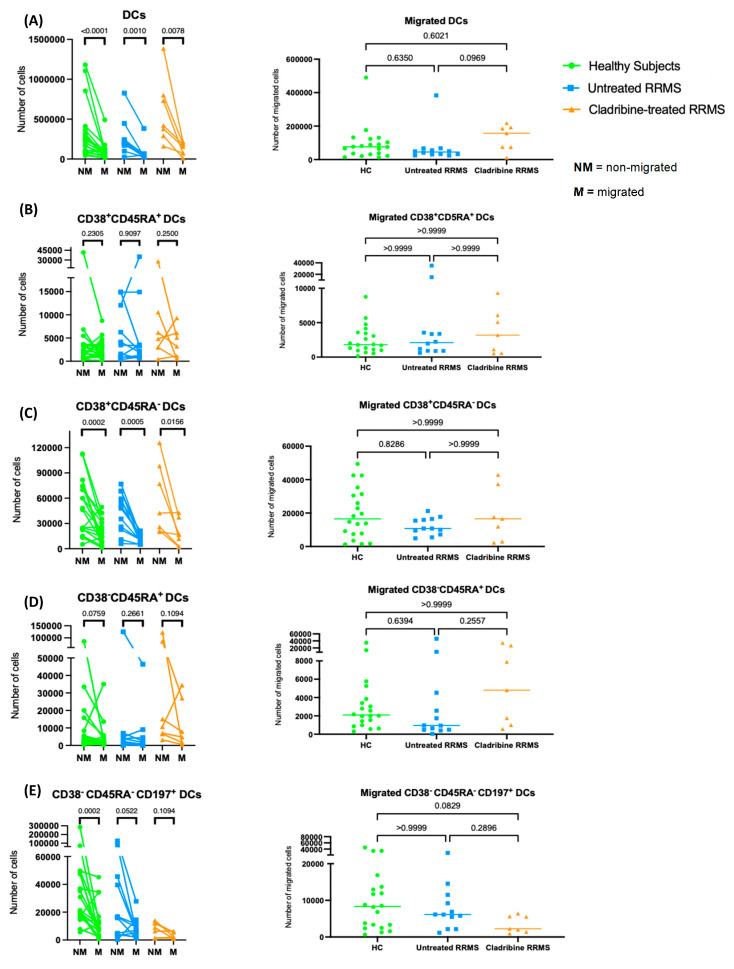
Cladribine does not alter DC transmigration. PBMCs from healthy subjects (*n* = 20), untreated RRMS patients (*n* = 12) and cladribine-treated RRMS patients (*n* = 7) were added to a transmigration assay and left overnight to migrate across the stimulated endothelial layer. (**A**) Absolute numbers of non–migrated (NM) vs. migrated (M) DCs in each group (**left**) and absolute numbers of migrated DCs in each group (**right**). (**B**) Absolute numbers of non–migrated (NM) vs. migrated (M) CD38^+^CD45RA^+^ DCs in each group (**left**) and absolute numbers of migrated CD38^+^CD45RA^+^ DCs in each group (**right**). (**C**) Absolute numbers of non-migrated (NM) vs. migrated (M) CD38^+^CD45RA^−^ DCs in each group (**left**) and absolute numbers of migrated CD38^+^CD45RA^−^ DCs in each group (**right**). (**D**) Absolute numbers of non–migrated (NM) vs. migrated (M) CD38^−^CD45RA^+^ DCs in each group (**left**) and absolute numbers of migrated CD38^−^CD45RA^+^ DCs in each group (**right**). (**E**) Absolute numbers of non–migrated (NM) vs. migrated (M) CD38^−^CD45RA^−^CD197^+^ DCs in each group (**left**) and absolute numbers of migrated CD38^−^CD45RA^−^CD197^+^ DCs in each group. Wilcoxon matched–pairs signed–rank test for M vs. NM. Kruskal–Wallis nonparametric one–way ANOVA with Dunn’s multiple comparisons test for migrated cells. The significance was dictated by *p* < 0.05.

**Figure 4 ijms-24-06487-f004:**
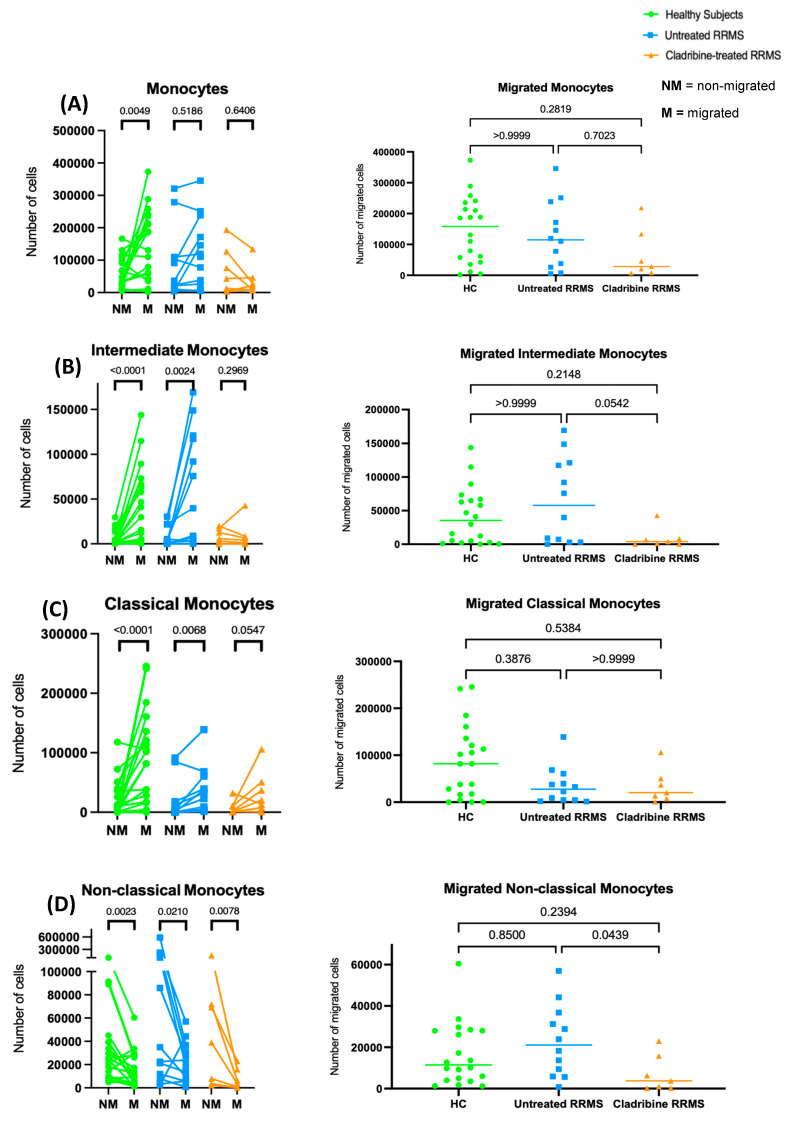
Cladribine impairs intermediate monocyte (CD14^+^CD16^+^) transmigration in RRMS patients. PBMCs from healthy subjects (*n* = 20), untreated RRMS patients (*n* = 12) and cladribine–treated RRMS patients (*n* = 7) were added to a transmigration assay and left overnight to migrate across the stimulated endothelial layer. (**A**) Absolute numbers of non–migrated (NM) vs. migrated (M) monocytes in each group (**left**) and absolute numbers of migrated monocytes in each group (**right**). (**B**) Absolute numbers of non–migrated (NM) vs. migrated (M) intermediate monocytes in each group (**left**) and absolute numbers of migrated intermediate monocytes in each group (**right**). (**C**) Absolute numbers of non–migrated (NM) vs. migrated (M) classical monocytes in each group (**left**) and absolute numbers of migrated classical monocytes in each group (**right**). (**D**) Absolute numbers of non–migrated (NM) vs. migrated (M) non–classical monocytes in each group (**left**) and absolute numbers of migrated non–classical monocytes in each group (**right**). Wilcoxon matched-pairs signed-rank test for M vs. NM. Kruskal–Wallis nonparametric one-way ANOVA with Dunn’s multiple comparisons test for migrated cells. The significance was dictated by *p* < 0.05.

## Data Availability

The data presented in this study are available upon request.

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
