# Peer review of "Oral Cladribine Impairs Intermediate, but Not Conventional, Monocyte Transmigration in Multiple Sclerosis Patients across a Model Blood-Brain Barrier"

_ijms, 2023, doi:10.3390/ijms24076487_

Round 1

Reviewer 1 Report

The manuscript by Lin et al. investigates the impact of oral cladribine on immune cell migration across the blood brain barrier (BBB) in relapse-remitting multiple sclerosis (RRMS) patients. While the effect of cladribine on lymphocyte migration is known, the effect on migration of monocytes and dendritic cells across the BBB is not. Therefore, using an in vitro model of the BBB, the investigators studied the transmigration abilities of monocytes and DCs using blood samples from healthy patients, untreated MS patients, and cladribine-treated MS patients.

Editorial changes should be applied to improve the manuscript; poor sentence structure, grammatical errors, and spelling mistakes detract from the clarity of the background information and discussion content. Also, all Greek symbols are shown as spiral icons, which need to be fixed. Aside from these issues, there are concerns with the precise language used to describe the results, the markers used to identify DC subsets, and labeling in the figures (see specific comments below).

Specific Comments:

1.     The title is misleading. While there is an association between oral cladribine and changes in cell migration, the investigations do not demonstrate that the effects observed in the cell migration assays are the result of inhibition by the drug.

2.     The wording used throughout the manuscript is also misleading. Some of the wording in the body of the manuscript implies that experimentation involved in the in vitro application of cladribine and assessment of its effects on cell numbers and migration.

3.     Were all patients in the treated cohort receiving cladribine at the time of blood collection? Were serum concentrations of cladribine measured in blood collected from patients in the treated cohort? This information could be very useful in constructing a hypothesis about the mechanism by which cladribine causes the observed effect.

4.     The manuscript should include some discussion on how cladribine is causing the effects observed in the migration assays. Even if blood samples collected from treated patients contained therapeutic concentrations of serum cladribine, steps used to isolate cells would have eliminated the drug from the cell preparations used in the BBB model experiments. Therefore, any changes in cell migration were consequences of indirect effects of the drug or changes in the cells that are sufficiently persistent to cause effects in the migration assays. The Discussion should address the potential indirect mechanisms by which the drug causes changes in cell migration.

5.     In the BBB model, cell confluence is not equivalent to the formation of a cell monolayer characterized by the formation of tight junctions and low permeability. The integrity of the monolayer in this type of assay is typically measured by other means (e.g., FITC-dextran transit across the cell layer or trans-epithelial electrical resistance).

6.     The authors need to fix Greek symbols throughout the paper

7.     More background needs to be provided on the mechanisms on the current drugs for MS and how they target the immune response. Also how do these drugs show the “key role of the immune system” during MS. 

8.     The Patients and Methods section does not describe the markers used to sort subsets of monocytes and DCs used throughout the study. It also does not list the antibodies used for flow cytometry. 

9.     In the results section 3.1, the marker used to identify subsets of DCs (CD38) is not specific to DCs. To my understanding, it is a marker for bone marrow progenitor cells (NK cells, monocytes, and lymphocytes), so how is it being used to specifically identify DCs? What is being done to ensure that DCs are being identified and not other cells that express CD38?

10.  Define what CD38-CD45RA-CD197+ DCs are and why they are significant to the study. Are they migratory? CD197 can indicate migration of DCs to secondary lymphoid organs.

11.  Line 231-233: Figure 2C is incorrectly used to explain the results of DC subsets in blood. These results are shown in Figure 2D.

12.  Figures 3 and 4: NM and M should be defined in the text and figure legends to clarify non-migrating and migrating cells.

13.  Figure 3 and 4: The authors should discuss the impact of blood sample availability on their results. Relative to the abundant number of samples available from healthy individuals, there were fewer samples tested from untreated RRMS patients and even fewer samples available from patients with treated RRMS. Does the number of samples from treated RRMS patients provide sufficient power to substantiate the conclusions?

14.  Since cladribine does not impact DC migration, the authors cannot conclude that the drug is globally “immune re-setting.”

Reviewer 2 Report

Dear authors, thank you for submitting an interesting article.

In the introduction part I would suggest to add some words about white matter as a ‘connection’ between two first sentences. Line 42 – correct INFg. Line 50 – repetition of ‘main’. The sentence "Adaptive immune cells are the main cells involved in MS pathogenesis whose main function is the eradication of foreign pathogens by direct cell killing of infected cells and by the secretion of cytokines, chemokines, or antibodies" may benefit from some clarification. It is not entirely accurate to say that killing pathogens by directly destroying cells is a mechanism found in the CNS in MS.  Or, did you simply want to mention the function of these cells in a state of health? In addition, I would specified ‘adaptive immune cells’. Moreover, it is overly simplistic to categorize monocytes and DCs solely as cells of the so-called 'innate' immune response, as they play a vital role in both adaptive and innate arms of the immune response. Line 77 – correct INF-b.

In general, I believe that the first three paragraphs of the introduction are written in a somewhat disorganized manner. The individual sentences need better integration, so that the entire text becomes more coherent and easily understandable, even for a novice reader.

Line 108 I would suggest rephrase the sentence.

The methodological part seems to be described adequately, but the question arises - were the exclusion criteria applied in the context of healthy volunteers (neurological diseases in the family or in the past, neurosurgery, recent infections)? I would add information about the number of people in each group in 2.1 paragraph.

Line 306 – high levels? What do you mean by that?

In addition, the study is limited by the small group of patients, especially those treated with oral cladribine. Of course, this is understandable and justified, but it makes reliable statistical assessment difficult. In my opinion, it is inadequate, on the one hand, to write in the summary about the need to assess how long the observed effect of cladribine on monocyte translocation to the CNS will last and, on the other hand, to claim that the study supports the thesis about the long-term effect of oral cladribine in MS patients. It is also worth remembering that this is only a model example created under artificially controlled conditions and care should be taken with the translation of this data to the clinic.

Round 2

Reviewer 1 Report

No further comments.

Reviewer 2 Report

Dear Authors,

Thank you for submitting the revised manuscript. I am glad to note that you have incorporated all the necessary corrections, which have had a positive impact on the quality of the data presented.

At this stage, I have no further comments or suggestions regarding the manuscript.

Regards!